# Exploring Perceived Stress in Mothers with Singleton and Multiple Preterm Infants: A Cross-Sectional Study in Taiwan

**DOI:** 10.3390/healthcare10081593

**Published:** 2022-08-22

**Authors:** Yu-Shan Chang, Yi-Chuan Cheng, Tsai-Chung Li, Li-Chi Huang

**Affiliations:** 1Department of Nursing, China Medical University Hospital, Taichung 406040, Taiwan; 2Department of Public Health, China Medical University, Taichung 406040, Taiwan; 3School of Nursing, Asia University, Taichung 413305, Taiwan; 4School of Nursing, China Medical University, Taichung 406040, Taiwan; 5Department of Nursing, China Medical University Children Hospital, Taichung 404333, Taiwan

**Keywords:** singular birth mothers, multiple birth mothers, social support in mothers with preterm infants, perceived stress in mothers with preterm infants

## Abstract

Objective: The aim of this study was to explore mothers’ perceived level of stress one month after hospital discharge following the birth of singleton and multiple preterm infants. Design: A cross-sectional design was used to compare mother’s perceived stress in two groups of postpartum mothers and the relationship of the theoretical antecedents and these variables. Setting: A neonatal intensive care unit in a medical center in Taiwan. Participants: Mothers of 52 singletons and 38 multiple premature infants were recruited. One month after the infant was discharged, the participants completed a self-reported questionnaire that included demographic data about the mother and infant, the 21-item Social Support Scale, and the 15-item Perceived Stress Scale. This was returned by email or completed at the outpatient unit. Analysis: Descriptive and inferential analysis. Results: The mean social support scores were 76.6 and 76.5 (out of 105) for mothers with singleton and multiple birth infants, respectively. The most important supporter was the husband. The mean perceived stress scores of 25.8 and 31.0 for mothers with singleton and multiple birth infants, respectively, were significantly different (*p* = 0.02). Sleep deprivation and social support were predictive indicators of perceived stress in mothers with preterm infants. Conclusions: We suggest that the differences in stress and needs of mothers with singleton and multiple births should be recognized and addressed in clinics. The findings of this study serve as a reference for promoting better preterm infant care.

## 1. Introduction

Pregnancy and giving birth result in significant psychological changes for women. The health of women and children has become a priority, as evidenced by the World Health Organization (WHO) declaration, “Make every mother and child count” [1]. Preterm birth is defined as the birth of a baby prior to a gestational age of 37 weeks. The WHO has reported an increase in the rate of preterm births, and studies have reported that there are over 15 million preterm infants born annually, accounting for more than 10% of all births [2]. In Taiwan, approximately 8–10% of infants are born preterm with higher numbers of babies in a pregnancy associated with an increased risk of preterm birth with 5.6% in singletons, 36% in twins, and 75% in multiple births [3,4]. Clinical psychologist Abidin put forward the concept of Parenting Stress Model in 1976. Parenting stress is the pressure that parents perceive and feel in the process of fulfilling their parenting roles and interacting with their children [5]. Specially, mothers with premature infants will face more pressure from changes in mother stability and changes in parental roles than mothers with full-term infants [6,7]. In the early postpartum period, mothers of preterm infants had higher levels of stress and depression than mothers of full-term infants, and maternal stress and depression were significantly correlated with weeks and weight of preterm infants [8,9]. Parents’ pervasive uncertainty, medical concerns and adjustment to the new parental role are concerns. Parents’ preparation for bringing the infant home is, thus, essential.

Stress is a crucial issue for mothers with preterm infants. Perceived stress is defined as the thoughts and feelings an individual has about how much pressure they are under at a given time [10]. Stress primarily results from an individual feeling that their life is uncontrollable and unpredictable, and from social events that carry an emotional burden. High levels of stress can result in negative physical and mental impacts, such as decreased cognitive abilities, distracted attention, poor memory, physical fatigue, insomnia, and feelings of burnout [11]. In 75% of mothers of preterm infants, the greatest difficulties and stress occur around one month after being discharged; this can affect the establishment of the parent–child relationship, which may then influence the infants’ development [12].

In societies, mothers are regarded as the primary caregivers of infants, and preterm infants are also responsible for nurturing and caring after they are discharged from the hospital. Many mothers report that they suffer from severe sleep disturbances [13,14]. Ohayon et al. (2017) [15] defined poor sleep quality as meeting at least one of the following requirements: onset of sleep latency > 31 min; awakening four or more times per night; and being awake after sleep onset > 51 min, sleep efficiency < 65%. When sleep cannot meet personal needs, it often causes headaches, shoulder and neck pain, fatigue, emotional agitation, which in turn affects cognition, and decision-making. As a consequence, this can even affect emotional behavior and cause future psychological problems, such as depression [16]. The American Academy of Sleep medicine recommend that adults should sleep seven or more hours per night to promote optimal health [17]. Sleeping less than 6 or more than 8 h significantly increases the risk of mortality; in terms of population and prevalence, sleep deprivation is more common and more likely to affect health than excessive sleep [16].

Social support includes the emotional, tangible, informational, and companionship support that an individual receives through interactions with family members, friends, and other support networks [18]. Social support works as a mediator for the stress response [18,19]. When individuals experience stress, social support allows them to feel connected to society. This can help increase their self-esteem, emotional stability, and sense of control [20]. It can also reduce the physiological and mental burdens caused by stress and meet the individual’s interpersonal needs [18,19]. Thus, higher levels of social support increase an individual’s capacity to deal with mental stress, reduce physiological problems, and increase the likelihood that the individual will be able to interact with and contribute to society [20,21].

The birth of multiple infants is exciting yet stressful. Twins require the primary caretakers to deal with twice the workload as for a singleton infant, which can result in increased anxiety while adapting to the new roles [20]. This can reduce optimal care and bring increased risk of the infants developing health problems or other complications. Moreover, the caregivers of twins are more likely to feel tired and frustrated because of failing to fully satisfy all the infants’ needs, increasing the risk of postpartum depression [22]. 

Most studies have focused on how parents adjusted to the birth of singleton preterm infants and coped with stress after discharge [12]. There has been little research on the responses of mothers of multiple preterm infants and whether these differ from those of mothers of singleton preterm infants. Thus, the aim of this study was to understand the levels of perceived stress received by mothers of singleton and multiple preterm infants at one month after the children were discharged. 

## 2. Methods

### 2.1. Design

This was a descriptive, cross-sectional study using questionnaire. Mothers of preterm infants were contacted via phone or clinical follow-up one month after their infants were discharged from the hospital.

### 2.2. Participants

The participants were mothers who gave birth to singleton or multiple preterm infants at a medical center in central Taiwan. They were recruited by purposive sampling, with the following criteria: the infant was born with a gestational age of less than 37 weeks and the mother was the infant’s primary caregiver within one month of the infant’s discharged. Mothers who had a disability, cancer, or mental illnesses and whose preterm infants had a congenital disorder were excluded from study. 

G*Power 3.1. software (Aichach, Germany)was used to estimate the sample sizes based on a pilot study of 10 participants. The sample size for detecting differences in social support was 36 participants per group (effect size = 0.66, power = 0.85, and α = 0.05). The sample size for detecting differences in perceived stress was at least 17 participants per group (effect size = 0.9, power = 0.85, and α = 0.05). Thus, the optimal sample size was at least 36 mothers with singleton and multiple preterm infants each, for a total of 72 participants. The final total was 90 participants, 52 mothers with singleton preterm infants, and 38 with multiple preterm infants; the study had a power of 90% to yield a statistically significant result.

### 2.3. Measures

The study used a self-reported questionnaire in three scales. The first scale consists of demographic data that included both mothers’ and infants’ information. Sleeping quantity and quality were measured by mothers’ perception of sleeping more/less than 6 h and good/poor quality. The second scale was Perceived Stress Scale (PSS), a 15-item questionnaire modified from Cohen’s study [23]. Questions such as, “In the last month, how often have you felt that you were unable to control the important things in your life?” were provided. This included questions answered on a five-point Likert scale from “always” (4) to “never” (0). The higher scores indicated higher levels of stress in mothers caring for their infants.

The third scale was the Social Support Scale (MOS), modified from Sherbourne and Stewart’s study [24]. The question is designed in four support domains, including emotional and information support, tangible support, affectionate support, and positive social interaction. Questions such as, “Someone you can count on to listen to you when you need to talk” were provided. This 21-item answered on a five-point Likert scale from “always” (5) to “never” (1). The higher scores indicated greater social support. This scale also included three open-ended questions: “Which family member provides the most positive support when you are taking care of infant(s)?”, “Which family member causes the most negative impact when you are taking care of infant(s)?”, and “What type of support would you most need to receive while caring for your infant(s)?” 

The instruments were used with the permission of the copyright holders. The instrument’s readability, accuracy, and adaptability were verified by a panel of experts and a pilot study of 10 mothers. Face validity was determined by experts’ panel review, with a content validity index (CVI) value of 0.95. Reliability and internal consistency measured by Cronbach’s alpha was 0.95 in the pilot study and 0.93 in the final study for the Perceived Stress Scale; and 0.97 in the pilot study and 0.95 in the final study for the Social Support Scale. 

### 2.4. Data Collection and Analysis

Data were collected from March 2016 to November 2016. Mothers who gave consent were given questionnaires for completion via email or at the hospital. Data were compiled using SPSS version 22.0 for statistical analysis. Frequency distributions, *t*-tests, one-way ANOVA, and chi-square were used to summarize the demographic information and difference in two groups. Pearson’s product-moment correlations were used to examine the relationships between MOS and PSS. General linear regression was used to predict PSS of mothers in the two groups. The variable of infant gestational age was controlled in regression models. An alpha level of 0.05 was designed for statistically significant.

## 3. Results

### 3.1. Demographic Data

Table 1 showed that the mean age of 52 mothers with singleton preterm infants was 33.7 (SD = 4.2) years old and the mean age of the 38 mothers with multiple birth preterm infants was 33.9 (SD = 3.7) years old. More than half (56%) of the singleton preterm infants were first-born children. Of the 38 sets of multiple preterm infants, 84% (n = 32) were first-born children. The mean gestational age of the singleton preterm infants was 33 (SD = 2.9) weeks and the mean gestational age of the multiple birth preterm infants was 34 (SD = 2.3) weeks. The mean birth weight was 1897 (SD = 586) gm in singleton preterm infants and 2044 (SD = 434) gm in multiple birth preterm infants. The chi-square test and *t*-test were used to test the homogeneity of the mothers and preterm infants in two groups. Only gestational age was statistically different in the two groups (Table 1). 

The three open-ended questions were responded to by 46 (88.5%) mothers with singleton preterm infants and 36 (94.7%) mothers with multiple preterm infants. The answers from #1: “Which family member provides the most positive support when you are taking care of infants?” The frequent answer was husband (65%, n = 30) in the mothers with singleton preterm infants, as well as in the mothers with multiple preterm infants (61%, n = 22). Question #2: “Which family member causes the most negative impact when you are taking care of infants?” The most frequent answer was none (73.1%, n = 38) and mother-in-law (13%, n = 6) in the mothers with singleton preterm infants. Correspondingly, none (63.2%, n = 24) and mother-in-law (25%, n = 9) were the responses in mothers of multiple preterm infants. Question #3: “What type of support would you most need to receive while caring for your infant?” Answers were tangible support (41%, n = 19) and informational support (30%, n = 14) in the mothers with singleton preterm infants, and tangible support (64%, n = 23) and informational support (17%, n = 6) in the mothers with multiple preterm infants. 

### 3.2. Perceived Stress and Social Support in Mothers with Singleton and Multiple Births

In Table 2, the mean scores (total score 60) of Perceived Stress Scale are revealed, where perceived stress was higher in mothers with multiple preterm infants compared to mothers with singleton preterm infants (*p* = 0.02). The mean scores (total score 105) of the Social Support Scale were revealed, where the social support was a little higher in mothers with singleton preterm infants compared to mother with multiple preterm infants with no statistical significance (*p* > 0.05). The level of social support had negative association with the level of perceived stress in both groups (*p* < 0.05, not shown in the table). Higher support leads to less stress. 

### 3.3. Factors Related to the Perceived Stress in Two Groups

The analysis showed the factors related to the perceived stress of the mothers with singleton preterm infants were mother’s sleep time and quality, educational level, the mother’s perceived health status, education, assistance care infant by mother-in-law, and social support. The factors related to the perceived stress of mothers with multiple preterm infants included the mother’s sleep time and quality, assistance with caring for the infants, and social support. The factors related to both groups were the mother’s sleep time and quality, mother’s perceived health status, educational level, number of births, and social support (Table 3).

The General Linear Regression analysis showed that sleep quantity and quality and someone in assisting were the indicators of perceived stress in mothers with multiple preterm infants. The GLM resulted that sleep quantity and quality, and social support were prognostic indicators of perceived stress for the mothers with preterm infants. The gestational age was controlled in the Models (Table 3).

## 4. Discussion

To our knowledge, this is the first study to explore and compare perceived stress in mothers with singleton and multiple preterm infants discharged one month at home in Taiwan. The main results showed that mothers with multiple preterm infants had higher perceived stress than the mothers with singleton preterm infants. One of the interesting findings from the statistics is that mothers-in-law were mentioned as having a negative impact on caring for infants in the open questions and the related factor of perceived stress for mothers with singleton preterm infants, but not in the mother with multiple preterm infants. The sleep quantity and quality and social support were beneficial factors for perceived stress among the mothers with preterm infants.

The study explores the perceived stress and social support received by mothers of singleton and multiple preterm infants at one month discharged at home. The mean age of the mothers with preterm infants in this study was 33.7 years, which is in line with the age range of 30–34 years at which, on average, Taiwanese women give birth to their first child [25]. In this study, the mean gestational ages of singleton and multiple preterm infants were 32.9 weeks and 34.4 weeks, respectively. These values are comparable to Lee’s study, where the mean gestational ages of singleton and twin preterm infants were 32–33 weeks and 36–38 weeks, respectively [3].

### 4.1. Social Support of Mothers with Singleton and Multiple Preterm Infants

Our study showed that both groups of mothers received social support in a moderate level (76 score out of 105). This might be a result of the postpartum confinement and one-month postpartum traditional care, whereby mothers would get more family support in Taiwan than in other countries [26]. Relying on tangible support and childcare advice of family members is typical in the Taiwanese social culture, where there is a strong emphasis on strong familial bonds [27]. For both groups of mothers, the categories that received the highest scores on the social support scale were having someone to take them if they needed to see a doctor and having someone to cook when they were unable to cook for themselves. Tangible support, such as providing money, spending time, improving surrounding environment, and doing chores, was the form of support the mothers most needed. This type of support can provide crucial social interactions and resources that reduce the stress of the mothers and increase the well-being of the infants [1].

Husbands provided the greatest social support for mothers with preterm infants in this study. Studies have shown that mothers adapt to their new roles more quickly and experience less stress when their husbands and other family members share in taking care of the infant. Thus, fathers of preterm infants can be considered the most important support for positive impact; they should, therefore, be included in health education before the preterm infants go home so they can provide more comprehensive support for the mothers.

Another important support, information support, was displayed in the outcomes. Many mothers of preterm infants lack the confidence and skillset for preterm infant care. Study reported that parenting support occurs by proving consistent advice, providing sufficient information about self-management for mothers, involving parents in discharge planning, and enabling clinical care consistency [28,29]. Health professionals, therefore, provide informational support in the early postpartum, and family members provide tangible and emotional support when the infants are brought home [1,27].

### 4.2. Perceived Stress of Mothers with Singleton and Multiple Preterm Infants

Mothers of multiple preterm infants had statistically higher perceived stress scores than the mothers with singleton preterm infants. Previous studies have reported that mothers with twins often feel anxious when adapting to their new roles, resulting in higher stress than experienced by mothers of singleton infants [20,30,31]; mothers with multiple preterm infants face severe emotional and physical burdens in the first year of having children [23,31]. Although parents learn childcare techniques during the infants’ hospitalization, applying these techniques at home can be completely different, which would lead to increased anxiety, resulting in changes to family functioning [27].

The results of the open question revealed that mothers-in-law help had a negative impact on infant care. Generational gaps in childcare philosophy, parenting perspectives, and childcare skills all added to the tension between the new mothers and their mothers-in-law [32,33]. In contrast, our results show that mothers of multiple preterm infants experienced no increase in perceived stress when their mothers-in-law took part in childcare. This might be because additional sets of hands were needed for caring multiple infants [34]. To optimize care, nursing professionals must pay attention to not only care for the mothers, but also the different generational perspectives and skillsets of other family members. Developing a home-centered childcare plan is needed that includes assessments of the skills and needs of family members in baby care assistance. Facilitating the participation of family members in this process will create a stronger social support network that relieves the mothers’ stress and provides better quality care for the preterm infant.

### 4.3. Beneficial Factors of Perceived Stress in Mothers with Preterm Infants

Insufficient and poor sleep was the item in the Perceived Stress Scale that was most often selected by both groups of mothers in this study. In the present study, 65% of the mothers with singleton preterm infants and 82% of those with multiple preterm infants slept less than 6 h per night. This finding was consistent with Leonard and Denton’ study [30]. Mothers with multiple preterm infants lacked sleep because of the additional work and stress from looking after multiple babies. The first week after infants were discharged from the hospital, mothers could sleep for as little as 4 h per night and have poor perceived sleeping quality [35]. The danger of insufficient and poor quality of sleep can result in memory deterioration, loss of concentration, emotional distress, and even less positive parenting and health outcomes [13,36,37].

Another reason of mothers’ insufficient sleep might be infants’ sleeping schedule and feeding needs at midnight. Culture is important here, as co-sleeping (sleeping side-by-side or in the same room) is very common in Taiwan. The sleeping environment would also affect both mothers’ and infants’ sleep. Coping strategies in sleep issues were recognized and recommended [38]. Parental sleep may ultimately prove to be a useful intervention for promoting positive parenting [37]. Useful tips include developing a good bedtime routine, transitional objects, listening to music at bedtime, and creating a tranquil, dark, quiet, and temperate environment [39,40]. Understanding these aspects can be incredibly useful in helping to create a healthy sleep routine, and this information can improve both the babies’ sleep and the mothers’ sleep. It is, therefore, recommended that nursing professionals educate mothers on infants’ sleep and stress the importance of mothers’ sleep when infants are discharged from the hospital.

## 5. Limitations

This study had several limitations. First, it was based on self-reported questionnaires; this might lead to the possibility of reporting bias because of social desirability. Second, the participants were recruited from a single hospital in a central Taiwan; thus, the results may not be generalized to all mothers with preterm infants in Taiwan. Third, cultural difference could be a bias in this group of participants; Asian families might experience more support/interference than Western families. Fourth, the excluded criteria might be part of the reason of the observed issues. Fifth, the study did not measure the psychological maternal status, which might reduce the validity of the results. Finally, a more rigorous design and a larger regional or national survey would enable to further assess social support and perceived stress in mothers with singleton and multiple preterm infants.

## 6. Conclusions

Social support was found to be a benefit indicator of perceived stress in both groups of mothers with singular and multiple preterm infants. Moreover, poor sleep problems were also displayed as important indicators of mothers’ perceived stress, especially among mothers with multiple preterm infants. There is no standard care plan for multiple preterm infants. Clinical teaching should aim to develop such a plan to support those mothers with multiple preterm infants. Fathers and other family members (such as mothers/mothers-in-law) provide important social support for mothers with preterm infants, so it is recommended that they are taught the necessary aspects of childcare to allow them to be more effective in reducing the stress of the mothers and increase the well-being of the infants. Finally, it is recommended that health professionals evaluate the mothers’ sleep quality as soon as possible after preterm infants are discharged from the hospital. Ensuring that the mothers of preterm infants have enough sleep allows for a more effective support system and improves the quality of life of their families.

## Figures and Tables

**Table 1 healthcare-10-01593-t001:** Demographic data.

Variable	Singleton (n = 52)	Multiple (n = 38)	t/χ^2^	*p*-Value
	M ± SD	M ± SD		
**Mother**				
Age			0.83	0.76
Under 30 years old	11 (21.2)	11 (28.9)		
31–35 years old	23 (44.2)	14 (36.8)		
36 years old above	18 (34.6)	13 (34.2)		
Parities			10.9	0.12
First	29 (55.8)	32 (84.2)		
2nd and above	32 (61.54)	6 (15.79)		
Job				
No	18 (34.6)	12 (31.6)		
Yes	34 (65.4)	26 (68.4)		
Education			1.79	0.18
High school	11 (21.2)	4 (10.5)		
Colleges	41 (78.8)	34 (89.5)		
Household income (month)			1.20	0.55
Less than 40,000 NTD	13 (25)	8 (21)		
40,001–70,000 NTD	20 (38.5)	19 (50)		
More than 70,001 NTD	19 (36.5)	11 (28.9)		
Assisting with infants’ care (multiple choice)			90.0	0.45
Husband	34 (65.4)	27 (71.1)		
Mother	30 (57.7)	19 (50)		
Mother-in-law	24 (46.2)	20 (52.6)		
Other	10 (19.20)	7 (18.4)		
Sleep quantity and quality			0.00	0.14
More than 6 h and good	16 (30.8)	7 (18.4)		
More than 6 h and poor	2 (0.04)	0 (0)		
Less than 6 h and good	19 (36.5)	14 (36.8)		
Less than 6 h and poor	15 (28.8)	17 (44.7)		
**Premature Infants**				
Gestational age	32.90 ± 2.94	34.4 ± 2.26	−2.53	0.01 *
Birth weight	1897.30 ± 586.43	2044.60 ± 434.89	−1.31	0.20
Birth weight at one month	3967 ± 886.27	4127.80 ± 670.62	−0.94	0.35
Days of births	67.30 ± 30.79	63.60 ± 22.88	0.21	0.84
Days after discharge	42 ± 15.45	43.58 ± 17.02	−0.49	0.63

* *p* < 0.05, Chi-Squared Test, *t*-test.

**Table 2 healthcare-10-01593-t002:** The social support and perceived stress in mothers with singleton and multiple births at one month after discharge.

Variables	Singleton (n = 52)	Multiple (n = 38)	*t*	*p*
	M ± SD	M ± SD		
Social support	76.6 ± 12.3	76.5 ± 13.4	0.02	0.96
Emotional and information support	28.6 ± 4.98	28.4 ± 5.03	0.25	0.83
Tangible support	19.8 ± 3.73	19.4 ± 3.97	0.54	0.59
Affectionate support	11 ± 2.44	11.3 ± 2.44	−0.53	0.59
Positive social interaction	11.2 ± 2.72	11.4 ± 2.27	−0.34	0.73
Perceived Stress	25.8 ± 11.07	31.0 ± 9.98	−2.30	0.02 *
	^a^ −0.44 *	^a^ −0.31 *		

* *p* < 0.05, *t*-test, ^a^: Pearson correlation.

**Table 3 healthcare-10-01593-t003:** The effect factors of perceived stress in mothers with preterm infants.

	Predict Model of Singleton (n = 52)		Predict Model of Multiple (n = 38)		Predict Model of All (n = 90)	
Variables	Coefficient	*p*	Coefficient	*p*	Coefficient	*p*
Intercept	54.16	0.000	26.39	0.22	52.92	0.000
Sleep quantity/quality (Ref: over 6 h and good)						
Less than 6 h and good	−1.67	0.64	11.51	0.002 *	2.15	0.42
Less than 6 h and poor	4.66	0.24	16.87	0.00 **	8.98	0.002 *
Over 6 h and poor	8.23	0.26			12.04	0.08
Perceived health status (Ref: Bad)						
Good	−3.45	0.27			−1.53	0.53
Education (Ref: Colleges)						
Senior high school	−3.62	0.28			−4.09	0.13
Assisting mother-in-law (Ref: Yes)						
No	−3.78	0.19				
Someone in assisting (Ref: Yes)						
No			15.84	**0** **.048 ***		
Number of births (Ref: multiple)						
Singleton					−4.18	0.06
Social support	−0.23	0.05	−0.068	0.486	−0.21	**0.016 ***
Gestational age	−0.23	0.28	−0.07	0.897	−0.29	0.11

* *p* < 0.05, ** *p* < 0.001, General Liner Regression.

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
