# Peer review of "Exploring Perceived Stress in Mothers with Singleton and Multiple Preterm Infants: A Cross-Sectional Study in Taiwan"

_healthcare, 2022, doi:10.3390/healthcare10081593_

Round 1
Reviewer 1 Report
Dear Ms. Kelly Li, Assistant Editor,
Manuscript Number: Manuscript ID: healthcare-1851409
Title: Exploring Perceived Stress in Mothers with Singleton and Multiple Preterm Infants
I carefully reviewed the manuscript which explores mother’s perceived stress one month after the birth of their singleton or multiple preterm infants. The study deserves attention because can contribute to develop better care plan in theperinatal time. However, at the present form the manuscript is not ready for publication: it needs some improvements.
I point out some suggestions below here:
1. Introduction
When stress variable is introduced (with only a definition) no mention has been done to a referred theory about stress: authors only reported a list of negative effects on the mother’s wellbeing. I suggested to point out in more details the theoretical hypothesis on the base of this work. A clarification about which is the stress hypothesis model will better contextualize the interest to measure mothers’ sleep and received social support, after hospital discharge following the birth of singleton and multiple preterm infants.
2. Method
The paragraphs Design and Participants are well described. The authors excluded mothers who had a disability cancer, or mental illnesses and whose preterm infants had a congenital disorder but, could the percentile of those problems be reported with respect to the total sample? This might help with one of the limitations: “social desirability”.
Regarding measures (2.3. Measures) I would suggest to insert at list one illustrative item for each of the applied scale: Perceived Stress Scale (PSS) and Social Support Scale (MOS), for giving a more precise idea about the requests.
3. Results
Regarding the text related to table 2, I suggest to delete information that are already inserted in the table. I also suggest. to explain the direction of the significant difference: substitute “this was statistically significant between the two groups” with, for example, “perceived stress was higher in mothers with multiple preterm infants compared to mothers with singleton preterm infants”.
Please, add in table 2 the information related to “The level of social support had negative association with level of perceived stress in both groups”. I do not appreciate the information between brackets: (… did not show in the table).
4. Discussion
In section 4.1. Social support of mothers with singleton and multiple preterm infants, I would appreciate the clarification between comments derived from statistical analysis and opinions extracted from the answers to the three open-ended questions of the Social Support Scale (MOS).
In section 4.2. Perceived stress of mothers with singleton and multiple preterm infants, the authors discussed the results by referring to studies which reported high level of anxiety, emotional and physical burdens in mothers with twins but no measures of maternal psychological wellbeing has been collected in this study. I would suggest, at least, in the list of limitations to the present study, a comment related to the lack of information regarding the psychological maternal status.
Reviewer 2 Report
The manuscript was very easy to read, and I have some comments.
1. (Affiliation) How about writing the affiliations fully for the two persons?
2. (Title) Title needs to contain other information, such as study design, year, country name, study place. It is becoming too concise now.
3. (Methods in Abstract) “Methods: descriptive and inferential analysis.” is hard to understand, and you need to elaborate about it. In addition, design and participants are written already, and the category “Methods” might be too broad.
4. (Throughout the manuscript) How about adding row numbers in the main-text?
5. (Throughout the manuscript) Your way of citing a reference is wrong. For example, “.1” should be replace with “[1].” . Please correct it.
6. (Introduction)  Could you explain why you restricted subjects to mothers with pre-term babies?
7. (2.1 designs in 2. Methods) “using survey” is an ambiguous. Please clarify it.
8. (Table1) “p値” should be corrected.
9. (Results) A regression analysis result in Table 3 that shows that a dummy variable for singleton-or-multiple birth is not statistically significant. It means that multiple birth is not related to perceived stress taking into account other factors, and you need to mention about it in Results and Discussion.
10. (Discussion) Could you summarize what was newly revealed in this study in the first paragraph of Discussion? You mention about a few points in discussion, and it is difficult to understand what is the novelty of the results.
Author Response
Please see the attached file, Thank you.

Round 2
Reviewer 1 Report
I think that authors made all the useful changes to improve thier manuscript. The new version looks better organised, balanced with respect to the information reported and easy to follow the results. In particular, the results and tables are now well organised. The findings are conclusive and can help to improve the clinical work in the parenting support, I belive that the manuscript is now ready for its pubblication.
Author Response
Thank you for your affirmation
Reviewer 2 Report
Thank you for the revision. I have some minor comments.
1. (Title)How about changing the title to “Exploring Perceived Stress in Mothers with Singleton and Multiple Preterm Infants:A Cross-Sectional Study in Taiwan”? There is no reason the authors hide the country’s name.
2. (Abstract)”Setting: A neonatal intensive care unit in a medical center” → ”Setting: A neonatal intensive care unit in a medical center in Taiwan”
